# Augmented Reality Surgical Navigation in Minimally Invasive Spine Surgery: A Preclinical Study

**DOI:** 10.3390/bioengineering10091094

**Published:** 2023-09-18

**Authors:** Xin Huang, Xiaoguang Liu, Bin Zhu, Xiangyu Hou, Bao Hai, Dongfang Yu, Wenhao Zheng, Ranyang Li, Junjun Pan, Youjie Yao, Zailin Dai, Haijun Zeng

**Affiliations:** 1Pain Medicine Center, Peking University Third Hospital, Beijing 100191, China; hay221@163.com; 2Department of Orthopedics, Peking University Third Hospital, Beijing 100191, China; hxy21hh@163.com (X.H.); haibao@bjmu.edu.cn (B.H.); 3Department of Orthopedics, Beijing Friendship Hospital, Beijing 100052, China; zhubin_ortho@163.com; 4State Key Laboratory of Virtual Reality Technology and Systems, Beijing Advanced Innovation Center for Biomedical Engineering, Beihang University, Beijing 100191, China; yudongfang1995@buaa.edu.cn (D.Y.); lry@haut.edu.cn (R.L.); 5Smart Learning Institute, Beijing Normal University, Beijing 100875, China

**Keywords:** augmented reality, minimally invasive spine surgery, real-time visualization, surgical navigation

## Abstract

Background: In minimally invasive spine surgery (MISS), where the surgeon cannot directly see the patient’s internal anatomical structure, the implementation of augmented reality (AR) technology may solve this problem. Methods: We combined AR, artificial intelligence, and optical tracking to enhance the augmented reality minimally invasive spine surgery (AR-MISS) system. The system has three functions: AR radiograph superimposition, AR real-time puncture needle tracking, and AR intraoperative navigation. The three functions of the system were evaluated through beagle animal experiments. Results: The AR radiographs were successfully superimposed on the real intraoperative videos. The anteroposterior (AP) and lateral errors of superimposed AR radiographs were 0.74 ± 0.21 mm and 1.13 ± 0.40 mm, respectively. The puncture needles could be tracked by the AR-MISS system in real time. The AP and lateral errors of the real-time AR needle tracking were 1.26 ± 0.20 mm and 1.22 ± 0.25 mm, respectively. With the help of AR radiographs and AR puncture needles, the puncture procedure could be guided visually by the system in real-time. The anteroposterior and lateral errors of AR-guided puncture were 2.47 ± 0.86 mm and 2.85 ± 1.17 mm, respectively. Conclusions: The results indicate that the AR-MISS system is accurate and applicable.

## 1. Introduction

Minimally invasive surgery is a significant trend in surgical development. How to make minimally invasive surgery more accurate is an urgent problem, especially for minimally invasive spinal surgery (MISS). The puncture positioning procedure is crucial in MISS; at present, it still relies on the guidance of X-ray fluoroscopy and the surgeon’s own experience [1]. In this procedure, there are problems, such as long learning curves, the inaccurate positioning of puncture points on the body surface, potential damage caused by puncture error, and frequent radiation exposure [2,3,4].

Although navigation and robot-assisted surgical techniques have been applied in clinical practice [5,6], they are not suitable for primary hospitals due to the expensive equipment needed and complicated requirements [7]. Moreover, an additional surgical incision for fiducial markers, such as spinous process bone clamps, is usually required [8], which cannot meet the requirement of minimized invasiveness.

Augmented reality (AR) is a technology that combines virtual information with the real-world environment. It is widely used in various fields, such as education and entertainment [9]. In the medical field, AR technology can be used to show information inside the body [10,11]. This is meaningful especially for MISS because unlike in open surgery, the surgeon cannot directly see the patient’s internal anatomical structure in MISS. The implementation of augmented reality technology may solve this major problem.

In our study, we combined augmented reality, artificial intelligence, and optical tracking to enhance the augmented reality minimally invasive spine surgery (AR-MISS) system. The AR-MISS system has three functions: AR radiograph superimposition, AR real-time puncture needle tracking, and AR intraoperative navigation. The purpose of this study was to evaluate the feasibility and accuracy of this novel technique by animal experiments.

## 2. Materials and Methods

### 2.1. Animal Preparation

All experiments were performed following the guidelines prescribed by the Institutional Animal Care and Use Committee (IACUC) under authorization No. A2019016. Twenty male adult beagles (Marshall, Beijing, China) were used as the experimental animals due to their suitable size, robust physical health, and clear spinal imaging. These beagles were 3 years of age and had an approximate weight of 10 kg. The animals were housed individually and fed with specific standard laboratory chow ad libitum.

The beagles were placed in a prone position. Preoperatively, general anesthesia was administered to the animals. After intravenous anesthesia induction with dexmedetomidine (0.008 mg/kg) and Zoleti (0.8 mg/kg), orotracheal intubation was performed. Anesthesia was maintained by continuous inhalation of 2% isoflurane. The anesthetic reaction was observed; if necessary, additional anesthetic could be added. Under anesthesia, the dorsal hair was shaved and prepared. The surgical site was disinfected thoroughly and draped.

### 2.2. System Components and Workflow

The AR-MISS system consists of a hardware system, a location kit, and self-developed software (version V1.0). The hardware system includes an infrared positioning device (Polaris Spectra, NDI, Waterloo, ON, Canada), two depth cameras (ZED Mini, Stereolabs, San Francisco, CA, USA), and a workstation with a monitor. The location kit includes custom-made noninvasive fiducial markers and a puncture needle locator. The system was implemented with a C-arm (Brivo OEC 715, GE, Boston, MA, USA) in the animal-specific operation room with lead protection (Figure 1). The workflow of the system is shown in Figure 2.

### 2.3. Augmented Reality Radiograph Superimposition

The AR-MISS system has an AR radiograph superimposition function. The noninvasive AR fiducial markers were placed on the animal’s back and flank in the horizontal direction and perpendicular direction via the radiolucent bracket (Figure 3A). The side length of the fiducial markers ranges from 55.0 mm to 66.5 mm. The anteroposterior (AP) and lateral radiographs were taken and exported to the AR-MISS system. Real intraoperative videos of the posterior and lateral positions were also simultaneously captured by two binocular depth cameras and transmitted to the system in real time (Figure 4A,B). To ensure consistent magnification ratios between the fiducial markers in the C-Arm and those in the video, it is crucial to position the light source of the C-Arm and the camera on the same side, ideally in the same location.

The fiducial markers could be visualized both in the radiographs and in the videos (Figure 3A,C). We trained a neural network based on YOLO-v3 to identify the non-invasive fiducial markers in both intraoperative videos and fluoroscopic images. For intraoperative video, we captured the image from the video at one frame per second and labeled the location of the markers in the image data to build the dataset of the markers in the intraoperative video. We used LabelImg to label each marker in the image data set and obtain the bounding box of each marker, and the data were stored in the corresponding label file. We divided the constructed image dataset and the corresponding label file into the training set, validation set, and test set. The training set data and validation data were input into the YOLO-v3 pre-training model for training. In the end, the trained model was utilized to detect and acquire the position of markers in the video (Figure 3B). The detection algorithm of non-invasive markers for fluoroscopic images is similar (Figure 3D). Following the recognition, the radiographs, and the real intraoperative videos could be automatically registered and matched to achieve AR radiograph superimposition. The real intraoperative videos and the superimposed AR radiographs were then shown on the screen (Figure 4). The transparency of the AR radiograph was set at 50%.

To evaluate the precision of the AR radiograph superimposition, 10 needles were fastened on the beagles’ backs and sides (Figure 5A). The positions of the needle tips in the AR radiographs were captured (Figure 5B). Radiographs were obtained to capture the real positions of the needle tips in the real radiographs (Figure 5C). We overlaid the AR radiographs (including real needles and AR spine) with the real radiographs (including needles and spine in the radiographs) by registering them through the spine, and the positions of the needle tips in the AR radiographs and in the real radiographs were compared (Figure 5D). The distance error of the needle tip and the angle error of the needle were measured. A total of 20 experiments were completed. For each group, 10 syringe needles in the AP radiograph and the lateral radiograph were evaluated.

### 2.4. Augmented Reality Real-Time Puncture Needle Tracking

The AR-MISS system can track the puncture needle in real time. The fiducial markers were first placed in different positions. In each position, the location information was collected by the depth camera and the optical tracker, and then spatial registration was performed to achieve correspondence between the two coordinates in the real intraoperative space and the video space. The locator and the retroreflective markers were affixed to the proximal end of the puncture needle; thus, the optical tracker can track the 6-DOF motion data of the puncture needle in the real intraoperative video. After self-adaptive calibration and coordinate transformation were obtained, the AR virtual puncture needle was finally able to be superimposed in the intraoperative video fused with fluoroscopy images.

To evaluate the precision of AR real-time puncture needle tracking, the puncture needle was placed on the body surfaces of the beagles’ backs and flanks. In this situation, the real puncture needle was captured by the camera and shown on the screen, while the AR virtual puncture needle was tracked and shown on the same screen in real time (Figure 6). The distance error and angle error between the real puncture needle and the AR virtual puncture needle were measured using Image-Pro Plus software (version 6.0, Media Cybernetics, Rockville, MD, USA). A total of 20 experiments were performed. For each group, the needle was evaluated in twenty different places (ten in the AP view and ten in the lateral view).

### 2.5. Augmented Reality Navigation Guided Puncture

With the realization of AR radiograph superimposition and AR real-time puncture needle tracking, the AR-MISS system can achieve AR real-time navigation.

After general anesthesia, AP and lateral radiographs were taken and superimposed on the beagle’s back and flank in the video. The beagle’s back was sterilized and covered by drapes. The puncture needle was tracked by the AR-MISS system, and the tracking accuracy was first verified on the body surface. Then, the puncture needle was inserted into the beagle’s back. The direction of the needle was adjusted according to the relative position between the AR radiographs and the AR puncture needle in real time. Under the guidance of the AR-MISS system, the puncture needle was carefully advanced until the needle tip touched the vertebra.

A screenshot was taken, and the final positions of the AR needle in the AR radiographs were recorded. Then, the real radiographs were taken, and the final positions of the real needle in the real radiographs were recorded (Figure 7). The positions of the AR needle and the real needle were compared. The needle tip distance error and angle error were evaluated. A total of 20 experiments were performed.

### 2.6. Postoperative Management

After the operation, the beagles were housed continuously for one week. The overall situation and the motor function of the beagles were observed. In the first three days after the experiment, the beagles were administered subcutaneous meloxicam 0.1 mg per kg body weight daily for analgesia. After one week of observation, the beagles were anesthetized by isoflurane inhalation (concentration: 4%; the tidal volume: 10 mL per kilogram of body weight; the number of breaths: 20). The inhalation was maintained for 5 min to achieve deep anesthesia, and 10 mL of 10% potassium chloride solution was injected intravenously for euthanasia.

### 2.7. Statistical Analysis

Experimental data are expressed as the mean ± standard error (SE). All statistical computations were performed using SPSS software (version 18.0, SPSS Inc., Chicago, IL, USA).

## 3. Results

In the experiment, all the AR radiographs were successfully superimposed in the real intraoperative videos. The AR radiograph distance error distribution diagram and angle error distribution diagram are shown in Figure 8. The distance errors were distributed mainly from 0 to 2.0 mm, and the angle errors were distributed mainly from 0 to 2.6°. The average distance error of the anteroposterior AR radiographs was 0.74 ± 0.21 mm, the average distance error of the lateral AR radiographs was 1.13 ± 0.40 mm, the average angle error of the anteroposterior AR radiographs was 0.62 ± 0.54°, and the average angle error of the lateral AR radiographs was 1.05 ± 0.77°.

Before puncture, the puncture needles were tracked and shown in the real intraoperative videos successfully. The accuracy of the AR real-time puncture needle was evaluated by the tracking experiment. The tracking distance error distribution diagram and angle error distribution diagram are shown in Figure 9. The distance errors were distributed mainly from 0 to 2.4 mm, and the angle errors were distributed mainly from 0 to 3°. The AP tracking average distance error was 1.26 ± 0.20 mm, the lateral tracking average distance error was 1.22 ± 0.25 mm, the AP tracking average angle error was 1.87 ± 0.81°, and the lateral tracking average angle error was 0.47 ± 0.13°. In addition, the latency of the tracking procedure of tracking was approximately 0.2 s.

The puncture process was finally guided by the AR-MISS system. The results showed that the position of the AR puncture needle in the AR radiograph was similar to the position of the real puncture needle in the real radiograph (Figure 7). The AR-guided puncture distance error distribution diagram and angle error distribution diagram are shown in Figure 10. The AP puncture average distance error was 2.47 ± 0.86 mm, the lateral puncture average distance error was 2.85 ± 1.17 mm, the AP puncture average angle error was 0.87 ± 0.78°, and the lateral puncture average angle error average angle error was 3.54 ± 2.82°. In the experiment, no massive bleeding, nerve injury, or other adverse events were observed.

## 4. Discussion

Recently, AR has become one of the most promising technologies. Unlike virtual reality technology, AR can be used not only for preoperative planning but also for intraoperative interaction with the surgeon [12]. AR navigation is different from traditional navigation. The current navigation systems only show virtual surgical instruments in the medical image [13], which cannot directly correspond to real surgical scenarios. Since the navigation image is separated from the visualization of the surgical site, doctors need to switch the field of view between the navigation screen and the surgical site and analyze the relative position information by thinking on their own. With the application of AR technology, the three spaces (the real intraoperative space, the video image space, and the medical image space) can be combined, and doctors can see information about the three spaces directly at the same time without looking away from the surgical site [14].

Accurate registration of the three-dimensional coordinates of the three spaces is a necessary step of the AR navigation system. The current spinal navigation systems often require the fixation of a fiducial marker on the exposed spinous process [8]. For open surgery, this procedure utilizes the open approach itself, but for minimally invasive surgery, the additional incision does not meet the principles of MISS. The noninvasive AR fiducial marker in our AR-MISS system only needs to be placed on the body surface, and it can be clearly identified and located in the three spaces. The application of the artificial intelligence algorithm ensures the accuracy of recognition and registration.

The registration of the three spaces was verified by the AR radiograph superimposition experiment. The results showed that the positions of the needle tips in the AR radiographs and the real radiographs were similar, confirming the success of the registration. In addition, AR radiograph superimposition is a promising function. Fluoroscopy is widely used in MISS for preoperative and intraoperative localization [15]. With the help of AR radiography, the structure and position of the spine can be observed from the surface of the body directly, providing a significant cue for preoperative localization and intraoperative puncture. At present, there are some difficulties in AR navigation; for example, it is difficult to present depth information clearly [16], and 3D AR representation sometimes prevents a clear view of the surgical field [14]. The biplanar AR fluoroscopy method can solve these problems, and it is consistent with the usual observation habits of surgeons.

A percutaneous procedure is necessary for minimally invasive spinal surgery. Surgeons can obtain a full view of the instrument before it is inserted into the body. Once the instrument is inserted, the internal part of the instrument is invisible. Intermittent fluoroscopy is usually needed to locate the instrument. The AR real-time puncture needle tracking function could make the internal part of the instrument visible. The AR-MISS system combined the AR radiograph superimposition function and the AR real-time tracking function, allowing the surgeon to see the real information (body surface and the external part of the instrument) and the virtual information (AR radiograph and the AR virtual instrument). The surgeon could adjust the AR instrument toward the target shown in the AR radiograph in real time and complete the percutaneous puncture procedure under unintermittent guidance of the AR-MISS system. This AR guidance process did not require continuous fluoroscopy; only fluoroscopy after the puncture was needed to check the final position of the instrument.

Precision is a key point in the application of the AR-MISS system. The precision requirement of AR medical applications is higher than that of applications of AR in other fields, such as entertainment. In our study, precision of up to 1 mm was confirmed in the AR radiograph superimposition experiment and the AR tracking experiment. From these distribution plots, it is shown that the majority of errors are concentrated within an acceptable range, although some outliners do exist. In the AR navigation guided puncture experiment, there are some outliners 9 or 10 mm away from the target. These outliners are likely attributable to the movement or breathing of the beagle, shifts in the camera or system, etc., leading to significant changes in position. As the beagles were kept in the prone position during the experiment, the body exhibited minimal positional changes in the AP direction during respiration, whereas lateral positioning showed notable fluctuations in the respiratory cycle. Consequently, outliners in the lateral position are more prominent than those in the AP position. However, the precision remained at a good level, which could meet the needs of clinical use. In addition, the tracking procedure could respond quickly to avoid the puncture risk caused by system delay.

The advantages of using an AR-MISS system include the following: it can provide the surgeon with a real-time view for guidance without continuous fluoroscopy, making the puncture positioning procedure easier; there is no problem with hand–eye coordination; the cost is low, and no complicated equipment is needed; and it is easy to use and does not significantly change the surgical procedure. The limitations of the AR-MISS system and the study include the following: the system solely performs real-time tracking of the puncture needle, and the object is only localized once during the overlay of augmented reality X-ray images. Subsequently, the fiducial markers device must be removed to avoid obstructing the puncture procedure. Thus, slight movement of the body, such as breathing or slight movement caused by puncture, may affect the accuracy. Functions such as movement tracking and correction need to be developed in the future. The functions of the AR-MISS system were verified in animal experiments, and more clinical trials are needed for further evaluation.

## 5. Conclusions

In summary, we combined augmented reality technology, artificial intelligence technology, and optical tracking technology to enhance the augmented reality minimally invasive spine surgery (AR-MISS) system. The functions of the system (AR radiograph superimposition, AR real-time puncture needle tracking, and AR intraoperative navigation) were verified by animal experiments. The results showed that the AR-MISS system was accurate and applicable. Even though augmented reality is not currently widely used in clinics, with the development of technology, there is no doubt that the future of AR in spine surgery is bright.

## Figures and Tables

**Figure 1 bioengineering-10-01094-f001:**
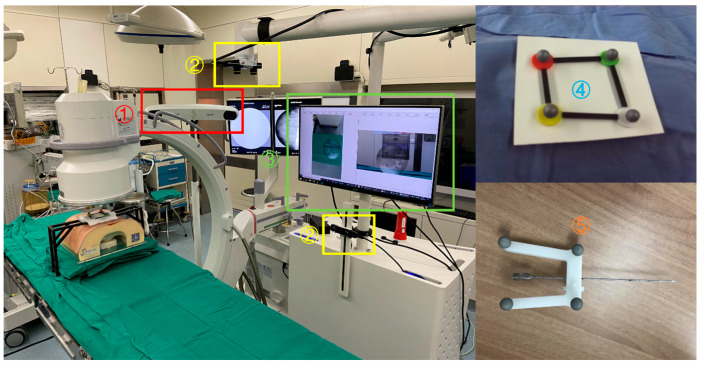
Composition of the augmented reality minimally invasive spine surgery system and its placement in the operating room. (**1**) NDI infrared positioning device; (**2**) frontal and lateral depth cameras; (**3**) workstation and display screen; (**4**) noninvasive fiducial markers; and (**5**) the puncture needle with a locator.

**Figure 2 bioengineering-10-01094-f002:**
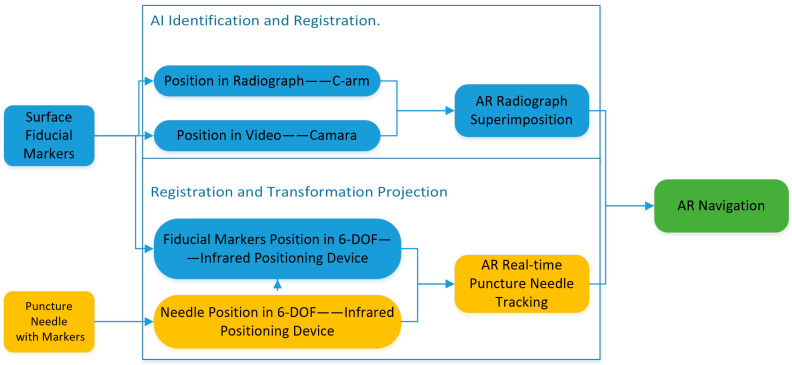
The workflow of augmented reality surgical navigation system.

**Figure 3 bioengineering-10-01094-f003:**
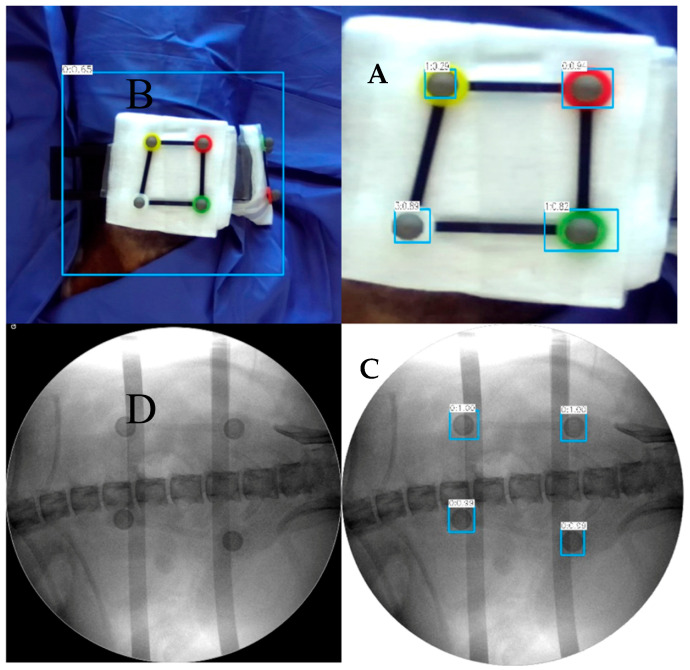
Noninvasive fiducial markers. The fiducial markers can be visualized in the videos (**A**) and the radiographs (**C**); the AR-MISS system could automatically recognize the fiducial markers in the videos (**B**) and the radiographs (**D**).

**Figure 4 bioengineering-10-01094-f004:**
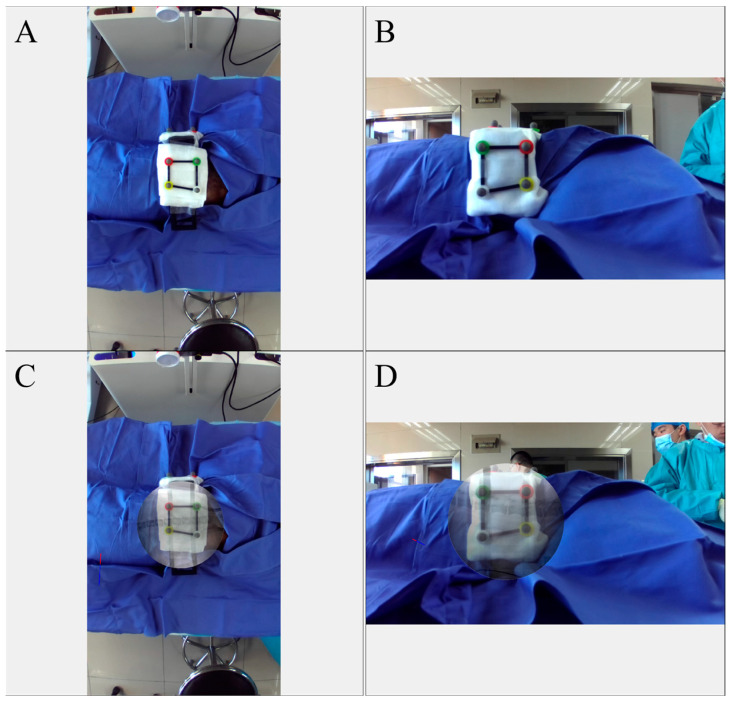
The real intraoperative videos before AR radiograph superimposition (**A**,**B**) and after AR radiograph superimposition (**C**,**D**). (**A**,**C**) anteroposterior view; (**B**,**D**) lateral view.

**Figure 5 bioengineering-10-01094-f005:**
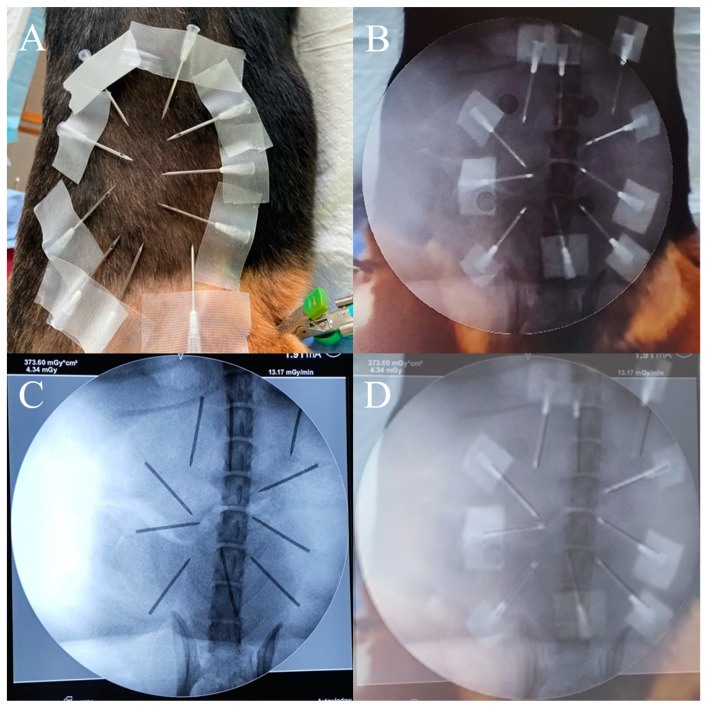
Accuracy evaluation for AR radiograph superimposition. (**A**) Ten needles were fastened on the beagle’s back. (**B**) The position of the needle tips in the AR radiograph. (**C**) The position of the needle tips in the real radiograph. (**D**) The fusion display to compare the position error.

**Figure 6 bioengineering-10-01094-f006:**
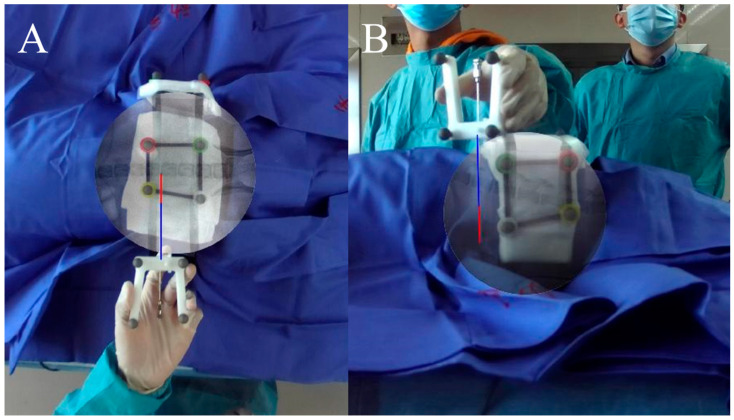
AR real-time puncture needle tracking. Before puncture, the puncture needle with a locator on the body surface was tracked, and the AR virtual needle (blue and red lines) was superimposed in the video in real time. The AR puncture needles nearly coincided with the real puncture needles. (**A**) anteroposterior view; (**B**) lateral view.

**Figure 7 bioengineering-10-01094-f007:**
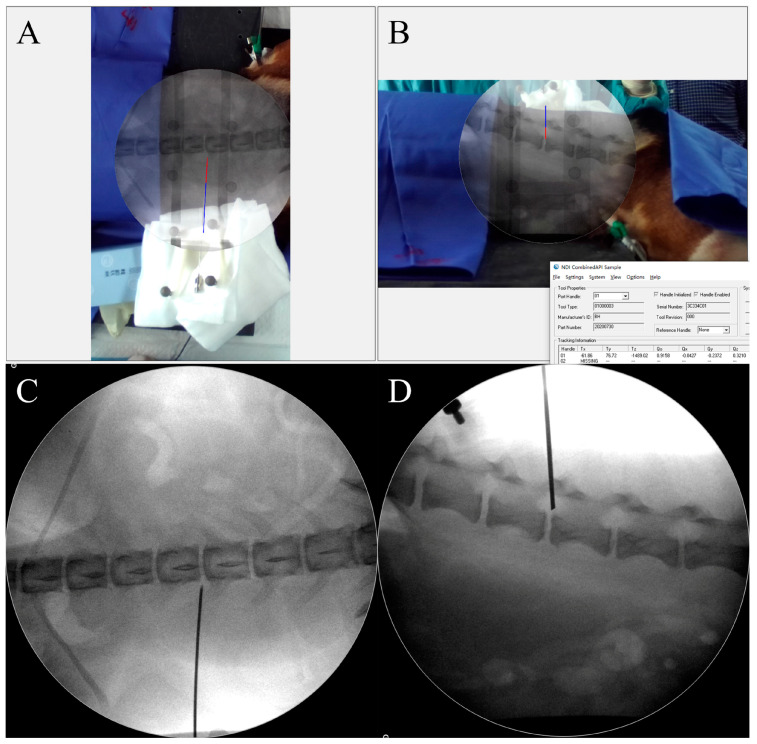
AR navigation guided puncture. (**A**,**B**) The position of the AR needle in the AR radiograph shown in the video; (**C**,**D**) the position of the real needle in the real radiograph shown in the C-arm. (**A**,**C**) Anteroposterior view; (**B**,**D**) lateral view.

**Figure 8 bioengineering-10-01094-f008:**
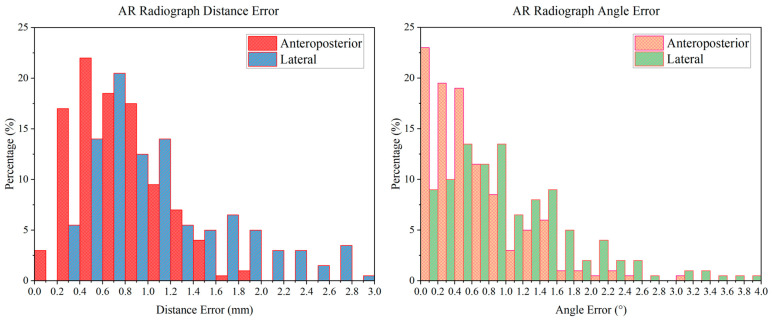
The error distribution diagram of AR radiograph superimposition. The distance errors were distributed mainly from 0 to 2.0 mm, and the angle errors were distributed mainly from 0 to 2.6°.

**Figure 9 bioengineering-10-01094-f009:**
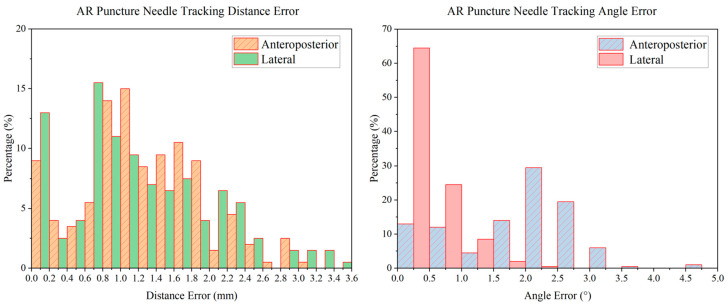
The error distribution diagram of AR real-time puncture needle tracking. The distance errors were distributed mainly from 0 to 2.4 mm, and the angle errors were distributed mainly from 0 to 3°.

**Figure 10 bioengineering-10-01094-f010:**
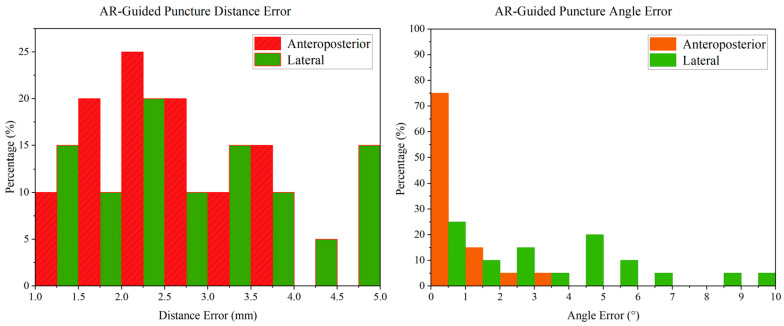
The error distribution diagram of AR navigation-guided puncture.

## Data Availability

Not applicable.

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
