# Peer review of "Augmented Reality Surgical Navigation in Minimally Invasive Spine Surgery: A Preclinical Study"

_bioengineering, 2023, doi:10.3390/bioengineering10091094_

Round 1

Reviewer 1 Report

The manuscript present an interesting approach to add AT guidance in mini-invasive spine surgery.  The hardware configuration presented is a state of the art set up for navigation. Their innovative contribution may rely in the deployment of a AI based software to track the segmented markers during the procedure. 

Honestly I have one big concerns: 

- The methods are not described at all. Nothing is told about the calibration of the C-Arm device to compensate for distortion, and nothing is described about the AI implementation. 

Without those details they just tested a system that I can not evaluate in terms of innovation or scientific sound. 

I strongly suggest the authors to completely rewrite the methods section actually describing the solution that has been tested. 

The manuscript needs strict English editing not only in terms of grammar but also in terms of style. 

Reviewer 2 Report

This well-prepared manuscript reported an animal study to quantify the accuracy of a newly developed Augmented Reality (AR) surgical navigation system (AR-MISS). The topic is interesting, the method is reasonable, and the results are convincing. 

My major critique is about the AI component in this manuscript. It is claimed that the AR-MISS system combined AR, AI, and optical tracking. The details of the AI application is lacking.  Please provide adequate description on  where AI is applied? What problem(s) be solved using AI? What algorithm/package/model were utilized? Training?  Prediction? Validation? etc. 

In addition, I offer the following comments, questions and suggestions:

L 45-46: missing proper transition sentence(s).

L59: what is the age of these young beagles?

L74: screen? howabout monitor, or 2D display?

 Figure 2 and related: Were to place the fiducial maker device? what is the size of fiducial markers? What is the fiducial markers are made of to ensure the radio-opaque?

Would you please provide a figure on the entire workflow?

L178-181: Figure 7, 8 and 9 need to be described in this paragraph/statistical analysis. Also, Following the distribution as presented in these figures, Mean and standard deviation may not be the proper parameter to quantify the accuracy. Furthermore, the accuracy ( how close to the target) and precision ( how close to each other)  are two different concepts. This study need to evaluate both.

Figure 9: please analysisand discuss the outliners. 9 or 10 mm away from the target need a n entire paragraph to discuss why?

L262:  High precision?  really?

L275-280: the limitations of this study: calibration whenever the movement occurs? Then, what the tracking for? Does this system only tracking the needle, not the object (animal)? What does the fiducial makers device for? 

L281-289: Conclusion: please revise accordingly. Also, does this system feasible? Does this study investigate the feasibility? If so, please provide aim, method, data and results to support this.

Reviewer 3 Report

The authors combined AR, AI, and optical tracking to improve the Augmented Reality System for Minimally Invasive Spine Surgery (AR-MISS). Results and debates are fascinating and beneficial for AR in spine surgery and with the help of AR X-rays and AR puncture needles, the puncture process could be visually controlled by the system in real time.  If the difficulties mentioned can be solved, this research can be published in Bioengineering.

  1. To give more information for the further implementation of this study, please give the reason why you chose beagles in Section 2.1, Animal Preparation. Is there any special reason for this, such as a condition that causes beagles' anatomy to become the ideal specimen?
  2. In line 239 of the discussion part, write, "The results showed that the positions of the needle tips in the three spaces were similar." Please provide more information to support this statement.
  3. In line 269 of the discussion section, it’s claimed that "In addition, the tracking procedure could respond quickly to avoid the puncture risk caused by system delay." When you claim it is quickly," how can you claim it? Please give an approximate time.
  4. In line 275 of the discussion section, there is a statement that "The limitations of the AR-MISS system and the study include the following: as the AR radiograph was superimposed on the body, slight movement of the body may have affected the accuracy, and functions such as movement tracking and correction need to be developed in the future. How can we categorize the movement as a slight movement? Gives more detail about it.

Round 2

Reviewer 1 Report

The manuscript has been improved and my comments addressed. 

Author Response

Comments and Suggestions for Authors

The manuscript has been improved and my comments addressed. 

Response: Dear reviewer, thank you very much for reviewing our manuscript and providing many helpful comments and suggestions, which will all prove invaluable in the revision and improvement of our paper, as well as in guiding our research in the future.

Reviewer 2 Report

Thanks authors addressed all questions I have. The revised version reads good. However, minor inconsistences exist. Please proofreading and modify.

For example, 

Line 198, using only mean +/- STE is inconsistent with Figure 8, 9 10. Please modify this paragraph following the figures. 

Line 282, while this paragraph starts with "precision" but discuss the "accuracy". Please modify to be consistent.

Also, the color scheme of Figure 8, 9, 10 looks inconsistent. Please double check, and modify as needed.

Reviewer 3 Report

I thank the authors for their replies, and the replies satisfy me. I would like to recommend the publication of the current manuscript in the journal.

Author Response

Comments and Suggestions for Authors

I thank the authors for their replies, and the replies satisfy me. I would like to recommend the publication of the current manuscript in the journal.

Response: Dear reviewer, thank you very much for reviewing our manuscript and providing many helpful comments and suggestions, which will all prove invaluable in the revision and improvement of our paper, as well as in guiding our research in the future.